# Mechanical Regulation of Oral Epithelial Barrier Function

**DOI:** 10.3390/bioengineering10050517

**Published:** 2023-04-25

**Authors:** Eun-Jin Lee, Yoontae Kim, Paul Salipante, Anthony P. Kotula, Sophie Lipshutz, Dana T. Graves, Stella Alimperti

**Affiliations:** 1Department of Biochemistry and Molecular & Cellular Biology, School of Medicine, Georgetown University, Washington, DC 20057, USA; eunjin.lee@nist.gov (E.-J.L.); yk729@georgetown.edu (Y.K.); sll111@georgetown.edu (S.L.); 2Microsystems and Nanotechnology Division, Physical Measurement Laboratory, National Institute of Standards and Technology, Gaithersburg, MD 20899, USA; 3Department of Chemistry and Biochemistry, College of Computer, Mathematical and Natural Sciences, University of Maryland, College Park, MD 20742, USA; 4Materials Science and Engineering Division, Material Measurement Laboratory, National Institute of Standards and Technology, Gaithersburg, MD 20899, USA; paul.salipante@nist.gov (P.S.); anthony.kotula@nist.gov (A.P.K.); 5Department of Periodontics, School of Dental Medicine, University of Pennsylvania, Philadelphia, PA 19104, USA; dtgraves@dental.upenn.edu

**Keywords:** epithelium, oral mucosa, microfluidics, mechanical stress, pressure, keratinocytes, organ-on-a-chip, fibronectin, barrier function

## Abstract

Epithelial cell function is modulated by mechanical forces imparted by the extracellular environment. The transmission of forces onto the cytoskeleton by modalities such as mechanical stress and matrix stiffness is necessary to address by the development of new experimental models that permit finely tuned cell mechanical challenges. Herein, we developed an epithelial tissue culture model, named the 3D Oral Epi-mucosa platform, to investigate the role mechanical cues in the epithelial barrier. In this platform, low-level mechanical stress (0.1 kPa) is applied to oral keratinocytes, which lie on 3D fibrous collagen (Col) gels whose stiffness is modulated by different concentrations or the addition of other factors such as fibronectin (FN). Our results show that cells lying on intermediate Col (3 mg/mL; stiffness = 30 Pa) demonstrated lower epithelial leakiness compared with soft Col (1.5 mg/mL; stiffness = 10 Pa) and stiff Col (6 mg/mL; stiffness = 120 Pa) gels, indicating that stiffness modulates barrier function. In addition, the presence of FN reversed the barrier integrity by inhibiting the interepithelial interaction via E-cadherin and Zonula occludens-1. Overall, the 3D Oral Epi-mucosa platform, as a new in vitro system, will be utilized to identify new mechanisms and develop future targets involved in mucosal diseases.

## 1. Introduction

Oral tissue homeostasis is governed by the function of various cell types, including perivascular cells, nerve cells, endothelial and epithelial cells, and their spatial arrangement in three-dimensional (3D) space through association with the extracellular matrix (ECM), providing biochemical and mechanical cues for cellular function. Specifically, alterations or disruptions of the epithelial barrier function interfere with homeostasis between the host and surrounding bacteria, contributing to inflammation, gingivitis, and periodontitis that cause tissue damage and may lead to tooth loss [1,2,3]. Despite the knowledge that bacteria-induced inflammation is involved in impairing epithelial function [4,5], the role of mechanical cues and tissue microenvironments in the regulation of barrier function in the oral mucosa is poorly understood.

The gingival sulcus is a V-shaped crevice between a tooth and the surrounding gingival tissue and is lined by sulcular epithelium. In 1959, Brill and Krasse [6] showed that fluorescein can be injected intravenously, and after 30 s, it is collected from gingiva pockets, demonstrating that the fluorescein passed the interstitial area through the blood stream and reached the epithelium. The major driving force is the hydrostatic interstitial pressure, which causes fluid filtration from gingival capillaries to the interstitial area and favors the formation of sulcular fluid. Specifically, sulcular fluid arises from a starling pressure gradient from the sulcus into the interstitial tissue area, indicating that epithelium is a permeable membrane and is exposed to mild mechanical stress [2,7,8,9]. In healthy states, this isotropic stress varies from 0.1 kPa to 2 kPa [2,10,11]. These measurements have been raised by integrating a wick (2 cm-long Teflon tube) into the subepithelial layer of the oral mucosa [12] and by 2 mm electrode strips, arranged in an orthogonal fashion and working as sensing points to measure the pressure distribution in the regions of interest [2,13]. In the presence of periodontal disease, the interstitial pressure increases, leading to leaky vessels, high gingival crevicular fluid flow, and, eventually, the formation of a periodontal pocket [14,15,16]. Thus, it is essential to investigate the role of mechanical cues in the regulation of oral tissue homeostasis.

To achieve this goal, the development of an in vitro system which will recapitulate the dynamics of the 3D microenvironment of the sulcus is essential. The utilization of animal models is often limited to investigating these 3D dynamic mechanisms, because they are expensive, laborious, and frequently fail to recapitulate human studies [17,18]. Thus, it is essential to develop and utilize 3D microphysiological in vitro systems, which will be used as alternatives to animal testing. Static microphysiological systems, including 3D microengineered tissues and organoids, have been widely used to dissect molecular mechanisms in development and disease pathogenesis [19,20]. However, they do not replicate organ-specific mechanical cues, such as flow and interstitial pressure, that are critical for accurate mimicry of tissue function in normal and disease states [4,18,21,22]. By contrast, microfluidic organ chips can provide all these features and mimic the dynamics of the 3D tissue microenvironment. Specifically, we developed a 3D microfluidic system to quantify the mechanical properties of microvascular barrier function in the presence of hydrostatic forces [23,24].

To achieve a deeper understanding and mimic the in vivo mechanical features of sulcular epithelium [11,12,25,26], we engineered an in vitro epithelial microfluidic model named the 3D Oral Epi-mucosa platform. The system integrates the mechanical components and the underlying matrix that oral keratinocytes are sensing. In general, the oral epithelium lies on a compact collagen fibrous tissue layer, namely the lamina propria, consisting of collagen and fibronectin (FN) [2,27,28,29,30,31]. Normal gingival tissue majorly contains type I collagen and, along with FN, provides structural integrity and mechanical strength to the connective tissue [32,33,34,35]. Importantly, when gingiva is injured, collagen and FN levels increase and are involved in mechanisms related to inflammatory responses (i.e., cytokines) and activation of immune cells [36,37]. Overall, COL1A1 and FN are essential for maintaining the structure and function of gingiva tissue, and they have been implicated in the regulation of adhesion, wound healing, inflammation, and periodontal tissue regeneration [32,33,34].

To this end, in this study, we evaluated the effect of a 3D fibrous microenvironment on epithelial barrier function by tuning collagen I and FN concentration; the application of mechanical stress (0.1 kPa) increased barrier function and the expression of key gene interepithelial adhesion molecules, such as E-cadherin. However, FN increased epithelial leakiness and decreased its expression. Overall, the development of a 3D Oral Epi-mucosa device as a new experimental model will permit finely tuned cell mechanical challenges and elucidate new pathways involved in oral diseases.

## 2. Materials and Methods

### 2.1. Cell Culture

Human immortalized gingival keratinocyte (HIGK) cells were generated by Jeffrey J Mans (University of Florida, Gainesville, FL, USA). The isolation of these cells and immortalization by retrovirus transduction of the HPV type 16, E6/E7 gene was described in [38]. These behave similarly to primary epithelial cultures and have been used extensively to characterize cellular and molecular events that occur in oral mucosal cells [38,39,40,41]. The cells were cultured in Keratinocyte Serum-Free Media (SFM) (Thermo Fisher Scientific, Waltham, MA, USA) supplemented with prequalified human recombinant Epidermal Growth Factor 1-53 (EGF 1-53) and Bovine Pituitary Extract (BPE) (Thermo Fisher Scientific). All experiments were performed with HIGKs in passage 4 to 5.

### 2.2. Formation of Collagen Gels

A solution of 3 mg/mL type I collagen (Corning, Bedford, MA, USA), 1 × M199 medium (Thermo Fisher Scientific), 1 mmol/L 4-(2-hydroxyethyl)-1-piperazineethanesulfonic acid (HEPES), 0.1 mol/L NaOH and NaHCO_3_ (0.035% by mass fraction) was polymerized for 1 h at 37 °C, as described previously [23]. Finally, 50 μg/mL FN (Col + FN) was added to a 3 mg/mL type I collagen solution (Table 1).

### 2.3. Fabrication of Microfluidic Platform

Molds for the microfluidic devices were created with a stereolithography (SLA) 3D printer (Form 3+; Formlabs, Soverville, MA, USA). Initially, computer-aided design (CAD) models were designed using 3D CAD, as demonstrated previously [24]. The exported standard triangle language (.STL) files were uploaded and were printed using clear resin (GPCL04; Formlabs), which is great for mold making. The uncured resin was removed from the surface of the 3D-printed molds by soaking and moving them in isopropyl alcohol (IPA) at Form Wash (Formlabs). Next, the washed molds were postcured at 405 nm light by using 13 multidirectional LEDS at Form Cure (Formlabs). Next, the cured molds were plasma treated for 5 min and silanized overnight in trichloro(1H,1H,2H,2H-perfluorooctyl)silane (Sigma, St. Louis, MO, USA). Polydimethylsiloxane (PDMS; Sylgard 184, Dow-Corning; Krayden, El Paso, TX, USA) devices were fabricated from these surface-modified molds. The PDMS devices were treated with 0.01% by volume fraction poly-L-lysine (PLL; Sigma) and 0.5% by volume fraction glutaraldehyde (Sigma) to promote collagen I adhesion. After washing overnight in water, steel acupuncture needles (diameter = 160 μm, Seirin, Kyoto, Japan) were introduced into the PDMS devices, and different collagen compositions were introduced and polymerized for 1 h at 37 °C. The following day, the needles were removed to create 160 µm-diameter channels within polymerized collagen gel, and a suspension of 10^6^ cells/mL HIGKs was introduced into the PDMS devices. The cells were adhered to the top surface of the channel for 2 min and then flipped to allow cells to adhere to the bottom surface of the channel for another 2 min. The nonadherent cells were washed out, and fresh media was replaced with the device. To apply mechanical stress, devices were placed on a pressure controller system for 8 h by applying a hydrostatic pressure of 0.1 kPa, as described previously [24].

### 2.4. Scanning Electron Microscopy

The collagen containers were simply fabricated using PDMS blocks and microscope cover glasses. To make a PDMS block (10 mm × 10 mm × 4 mm), the PDMS base and curing agent were mixed with a 10:1 ratio and poured into a petri dish. Air bubbles were removed from the PDMS mixture in a desiccator, and the PDMS was cured in a conventional oven for 6 h at 80 °C. The cured PDMS was cut into 10 mm × 10 mm blocks, and the center hole was drilled using a 5 mm-diameter punch. The microscope cover glass and PDMS block were permanently combined after activation with oxygen plasma. The collagen solutions (200 µL) were placed in the fabricated collagen containers and allowed to polymerize for 1 h at 37 °C. Then, SFM media was added on top of the polymerized collagen thereafter. The collagen containers were placed in an incubator overnight at 37 °C before performing a serial dehydration process with ethanol and hexamethyldisilazane (HMDS), as previously described [42]. Briefly, the collagen gels were fixed in 4% by volume fraction paraformaldehyde (PFA; Sigma) in phosphate-buffered saline (PBS) for 1 h at room temperature. The collagen gels were washed with PBS three times and with distilled water twice. The fixed collagen gels were dehydrated in 30%, 50%, 70%, 90%, and 100% double-distilled H_2_O/ethanol volume fraction series and were washed with 33%, 50%, 66%, and 100% ethanol/HMDS volume fraction series. Finally, the collagen gels were allowed to dry for 24 h on a glass slide. After the drying process, the collagen samples were mounted on an aluminum stud using conductive carbon tape. The samples were examined under scanning electron microscopy (SEM) (Zeiss, Hebron, KY, USA) with high vacuum mode with accelerating volt at 1 kV and magnificent of 2000× and 10,000×. The diameters were calculated using DiameterJ, which is a plugin for ImageJ. The resulting values were then used to create histograms.

### 2.5. Mechanical Testing

Shear modulus measurements were performed on a stress-controlled rheometer (MARS III, Thermo Fisher) using an 8 mm parallel-plate measurement geometry and a gap height of 1 mm. First, the polymerization temperature was set at 37 °C, and 1 mL of collagen gel solution was loaded into the rheometer. The outer edge of the sample was coated with a polyphenylmethylsiloxane (PPMS) oil to prevent evaporation, and the sample was held at temperature for 1 h prior to measurements. Angular frequency ω sweeps were performed with a strain amplitude of 0.02 rad/s in the range of 1 rad/s to 100 rad/s, and the stress and strain raw signals were monitored to confirm that the measurement was in the linear viscoelastic range (Appendix A). For all samples, measurement frequencies above 21 rad/s experienced significant nonlinearities and were not reported. The viscoelasticstress–strain response of the material was characterized by a complex shear modulus G* (ω) = G′(ω) + iG″(ω), where G′(ω) was the storage modulus and G″(ω) was the loss modulus. We denoted the magnitude of the complex modulus |G*| = √((G′)^2^ + (G″)^2^) at ω = 1 rad/s as the “shear modulus” in the Results section for brevity. The loss tangent tanδ = G″/G′ at ω = 1 rad/s was also reported. The uncertainty in the measurement of linear viscoelastic properties was 10%, and duplicate samples were measured to confirm reproducibility. The storage and loss moduli as a function of frequency were included in Appendix A.

### 2.6. Epithelial Permeability Measurement

To measure the epithelial barrier, as has been demonstrated previously [43,44,45], fluorescent dextran (70 kDa Texas Red, Thermo Fisher) was introduced into perfusion media (SFM) at a concentration of 12.5 µg/mL. The diffusion of the dextran was imaged in real time with a confocal microscope (LSM 800, Carl Zeiss, Dublin, CA) at 10× magnification. An image time sequence was analyzed by taking the mean intensity over regions next to the cell layer in successive images. The time derivative of the intensity was determined by linear regression for each region. The time derivative of the intensity, the mean intensity (*I*), and the capillary radius (*r*) were used to determine the diffusive permeability coefficient (P_d_) by the equation P_d_ = *dI*/*dt* × *r*/*2I*. The uncertainty was determined by the standard deviation of the measured permeability at the different locations, as described previously [23].

### 2.7. Immunofluorescence Staining

The 3D Oral Epi-mucosa microfluidic devices were fixed in 4% by volume fraction paraformaldehyde (PFA; Sigma) for 20 min at 37 °C, washed twice in PBS, permeabilized with 0.1% by volume fraction Triton X 100 in PBS for 2 h at room temperature, and treated with blocking solution (0.01% by volume fraction Triton X 100, 5% by mass fraction goat serum (Sigma) in PBS) overnight at 4 °C. 4′, 6-Diamidino-2-Phenylindole (DAPI) (1:300, Sigma) and Alexa Fluor 488 Phalloidin (1:200, Thermo Fisher) were incubated overnight at 4 °C. To capture cell–cell interactions, the cells were incubated with E-Cadherin (24E10) Rabbit mAb (1:100, Cell signaling) overnight at 4 °C. The next day, the cells were washed twice in PBS, and a secondary antibody (goat antirabbit IgG (H + L), Alexa Fluor 488 (1:100, ThermoFisher)), was added for 4 h at room temperature. After three times being washed, the devices were imaged using a confocal microscope (LSM 800, Carl Zeiss), and image analysis was carried out using ImageJ [46] by performing a maximum intensity z projection and merging the channels. For cell spreading and attachment assay, HIGKs were plated on collagen gels at the density of 1000 cells/cm^2^. After 4 h, the cells were fixed in 4% by volume fraction PFA (Sigma) for 5 min, washed twice in PBS, permeabilized with 0.1% by volume fraction Triton X 100 in PBS for 10 min, and treated with blocking solution (0.01% by volume fraction Triton X 100, 5% by mass fraction goat serum (Sigma) in PBS) for 1 h at room temperature. Next, the cells were incubated with DAPI (1:1000) and Alexa Fluor 488 Phalloidin (1:500, Thermo Fisher) for 1 h at room temperature. The cells were imaged using a confocal microscope (LSM 800, Carl Zeiss). Cell attachment and spreading areas were obtained using a cell profiler (https://cellprofiler.org/) (accessed on 1 January 2023) [47]. More than 80 cells were measured in three separate experiments.

### 2.8. RT-PCR and RT-qPCR Analysis

Total RNA was isolated from the 3D Oral Epi-mucosa platform using TRIzol Reagent (Invitrogen, Carlsbad, CA, USA) following the manufacturer’s instruction. SuperScript III cDNA Synthesis kit (Thermo Fisher Scientific) was used for reverse transcription polymerase chain reaction (RT-PCR) in an Applied Biosystems Veriti 96-Well Thermal Cycler. The quantitative real-time (RT)-qPCR was performed using a PowerTrack SYBR Green Master Mix and ViiA 7 Real-Time PCR System with a 96-Well Block (Applied Biosystems, Waltham, MA, USA), in accordance with the manufacturer’s instructions. The primers used in this study are described in Appendix A. The expression level of each gene was normalized to the expression level of 18S internal control. The relative gene expression was calculated by the standard curve method using the target Ct values and the Ct value for 18S.

### 2.9. Statistical Analysis

Statistical analysis of the data was performed using one-way analysis of variance (ANOVA), using Tukey’s post-test for more than two variables. The *p*-value was set to be significant if <0.05 unless stated otherwise in the text. All results were expressed as mean plus or minus one standard deviation. In each test, the number of independent experiments (N) was more than three, and the number of data points (n) in each experiment was different. Both N and n were shown in the figure legends.

## 3. Results

To investigate the role of the transepithelial fluidic dynamics of the oral sulcus [11,12,25,26] (Figure 1a), we developed a 3D microfluidic system named the 3D Oral Epi-mucosa platform. The microfluidic device was fabricated by 3D-printed scaffold, in this case by casting a single, hollowed cylindrical channel (160 µm diameter) into collagen I matrix within a PDMS mold containing a bulk chamber hosting the tube and reservoir chambers for introducing media into the system (Figure 1b). The device was connected to a pressure control system, and 0.1 kPa hydrostatic pressure was applied for 8 h (Figure 1c). The cells were attached to the device and organized into an engineered microepithelial tube, as presented in Figure 1d. Under those conditions, this system was applied to measure the epithelial barrier function by capturing the extravasation of fluorescently labeled 70 kDa dextran (Appendix A).

Initially, we investigated how the 3D microenvironment, which the cells are sensing, controls the barrier function. To this end, we modulated the stiffness of the underlying matrix by tuning the concentration of collagen from 1.5 mg/mL to 6 mg/mL. SEM images demonstrated that the number of fibers with a diameter of less than 60 nm at 6 mg/mL is the highest compared with the other two conditions (Figure 2a). In addition, the total stress response via the shear modulus demonstrated that collagen I gel at a concentration of 6 mg/mL showed approximately 9 times higher stiffness (*p* < 0.05) compared with collagen gel at a concentration of 3 mg/mL and approximately 45 times higher stiffness (*p* < 0.05) compared with collagen gel at a concentration of 1.5 mg/mL (Figure 2b). However, the loss tangent in these viscoelastic materials was independent of the collagen I concentration, indicating that the elastic contribution to the stress dominates over viscous dissipation (Figure 2c). Under these conditions, we evaluated the barrier function by dextran assay. The cells were attached within the device containing the different collagen I concentration and organized into the microepithelial tubes as demonstrated by the immunofluorescence staining in Figure 2d. The cells lined on 3 mg/mL collagen I gel demonstrated approximately 39 times and approximately 23 times decrease in epithelial leakiness (P_d_) compared with the cells lined on 1.5 mg/mL and 6 mg/mL collagen I gels, respectively (Figure 2e,f). Our results indicate that the mechanical properties of the 3D extracellular matrix tune the epithelial barrier function.

Epithelial barrier function is controlled by adhesion molecules that play an important role in intercellular interaction [48,49]. Herein, we investigated how the mechanical properties of the 3D ECM matrix regulate adherens and tight junction molecules. Our results demonstrate that cells lying on the intermediate substrate of 3 mg/mL compared with the soft substrate of 1.5 mg/mL expressed approximately 14 times higher junctional E-cadherin. Similarly, the cells lying on stiff substrates of 6 mg/mL inadequately expressed junctional E-cadherin compared with those lying on 3 mg/mL (Figure 3a,b). Moreover, gene expression analysis by RT-qPCR for genes involved in cell–cell contact and cell–matrix contact demonstrated that cells placed on an intermediate stiffness of 3 mg/mL expressed an increase in E-cadherin (CDH1) (≈1.7 times; *p* < 0.05), Zonula occludens-1 (ZO1) (≈1.6 times; *p* < 0.05), Occludin (≈3.1 times; *p* < 0.05), Integrin beta-1 (IΤGB1) (≈3.7 times; *p* < 0.05), and Integrin alpha-6 (ITGA6) (≈5.8 times; *p* < 0.001) compared with cells placed on a soft matrix of 1.5 mg/mL. Similarly, cells placed on intermediate stiffness of 3 mg/mL expressed an increase in CDH1 (≈5.78 times; *p* < 0.01), ZO1 (≈6.6 times; *p* < 0.01), Occludin (≈3.8 times; *p* < 0.05), IΤGB1 (≈674 times; *p* < 0.001), ITGA6 (≈22.9 times; *p* < 0.001), and Lamin5 (≈1.9 times; *p* < 0.05) compared with cells placed on a hard matrix of 6 mg/mL (Figure 3c). Taken together, these data demonstrate that the intermediate stiffness of the 3D matrix maintains the epithelial barrier via the upregulation of gene expression of cell adhesion molecules.

FN is one of the major components of the ECM oral mucosa [50]. Herein, we investigated how FN controls the mechanical properties of 3D ECM and, further, how those properties affect barrier function. Initially, we tested how the presence of FN at low concentration (5 μg/mL) (denoted as Col + FN(5)) and high concentration (50 μg/mL) (denoted as Col + FN(50)) (Table 1) in the 3D gel may affect the gel local topography and mechanical properties. SEM images demonstrated that fiber intensity for Col + FN(50) is higher compared with other conditions (Figure 4a). The topological differences of those gels were interconnected with the mechanical properties of the gels. Specifically, Col + FN(50) gels had approximately three times higher shear modulus compared with Col gel, indicating that Col + FN(50) gels are stiffer compared with Col (Figure 4b). However, FN did not have any effect on the loss tangent of these viscoelastic materials (Figure 4c).

Next, we evaluated whether the high concentration of FN controlled cell adhesion and spreading. To this end, HIGKs at 1000 cells/cm^2^ density were plated on the gels, and after 4 h, the cell attachment and spreading were examined (Figure 4d). Cell attachment on Col + FN (50) was approximately six times higher compared with Col (Figure 4e). Moreover, the cells on Col + FN(50) gels adopted a spindle-like morphology, and the cell spreading area was approximately three times higher compared with cells cultured on Col gels (Figure 4f). These results suggest that FN controls the local microenvironment that cells are sensing and, eventually, tunes the cell attachment and spreading.

As a next step, we investigated the role of a high concentration of FN (50 μg/mL) in barrier function. The cells were plated and attached within the device in the presence of FN (50 μg/mL) (denoted as Col + FN). Our immunofluorescence staining results show that in the absence of stress, the cells lined into the Col + FN matrix demonstrated approximately four times higher migratory phenotype and dropped their compact microepithelial tube formation compared with Col gels. However, the presence of stress dramatically reduced the migration of the cells by three times in the Col + FN matrix (Figure 5a,b). In addition, the presence of FN increased epithelial leakiness by approximately 30 times compared with the epithelial cells embedded in Col gels (Figure 5c,d), indicating that FN reversed the barrier integrity. Interestingly, the absence of mechanical stress enhanced the epithelial leakiness by approximately 17 times for the cells lined in Col + FN gel and by approximately 10 times for cells lined in the Col, indicating that the absence of mechanical stress and the presence of FN adversely regulated the epithelial integrity (Figure 5d).

The profile of genes involved in the regulation of barrier function showed that the absence of mechanical stress and the presence of FN decreased junctional E-cadherin localization (Figure 6a,b and Appendix A). Similarly, gene expression analysis showed a decrease in CDH1 (≈1.8 times; *p* < 0.05), ZO1 (≈9.5 times; *p* < 0.05), Occludin (≈1.6 times; *p* < 0.05), IΤGB1 (≈2 times; *p* < 0.05) and ITGA6 (≈5.8 times; *p* < 0.01) in the absence of stress (Figure 6c). Unexpectedly, mechanical stress had the minimum effect on Lamin5. Taken together, these data demonstrate that the presence of mild mechanical stress induced an epithelial barrier, while the presence of FN in the underlying matrix disrupted it.

## 4. Discussion

A deeper understanding of oral pathophysiology requires the development of 3D culture systems that can recapitulate the oral ECM structures, mechanics, and complex cell–cell and cell–matrix interactions that occur in vivo [51,52,53]. Recent studies have demonstrated the development of 3D microfluidic platforms that have been widely used in oral, dental, and craniofacial areas [54]. Specific efforts related to oral mucosa include the Gingival Crevice-On-Chip platform, which recapitulates host–oral microbiome interactions [55]; oral mucosa-on-a-chip, which has been applied to evaluate the responses of bacteria to dental materials [56]; and oral mucositis-on-a-chip, which investigates the role of chemo- and radiation treatments in oral tissue [56]. In our study, we developed the 3D Oral Epi-mucosa platform to recapitulate the mechanical forces generated by hydrostatic components and the underlying matrix that the epithelial cells are sensing. The application of this platform permits the measurements of epithelial integrity, avoiding the limitations of animal studies and static microphysiological systems [57]. Moreover, we were able to investigate key mechanobiological targets involved in epithelial barrier function, which may propagate new treatments for oral diseases, such as periodontitis.

Despite the fact that gingival mucosa is exposed to interstitial forces in healthy and diseased cases (i.e., periodontitis) [4,5,11,12,25,26], mechanical-based mechanisms that modulate barrier function are not well understood [58,59,60,61,62,63]. Using the 3D Oral Epi-mucosa platform, our results show that low-level interstitial pressure enhances the epithelial barrier via consistent changes in E-cadherin and is disrupted by the presence of FN. We demonstrated that these mild mechanical forces (0.1 kPa) improve barrier function and redistribute junctional E-cadherin that has been involved in regulating barrier function [5,64,65,66,67]. Studies have shown that the response of cells to forces is mediated via dependent cell–cell interactions and involves the regulation of cadherins [68,69,70]. This agrees with our studies, where we showed that the presence of stress enhances intercellular interaction via localization of E-cadherin in the junctional area and the upregulation of tight junction molecules, such as ZO1 and Occludin.

There are significant differences in the biomechanics of periodontal tissues in nonperiodontitis and periodontitis patients. Several studies have reported that periodontitis-infected tissue has demonstrated a reduction in stiffness and tensile strength and an increase in porosity compared with healthy tissue due to the destruction of collagen fibers in the periodontal ligament, which is the load bearing structure in the periodontium [71,72,73,74,75]. Despite this, much less is known about the oral epithelial barrier in response to the mechanical properties of the matrix. In this study, the mechanical properties of the matrix at a range of 1.5 mg/mL to 6 mg/mL collagen concentrations led to the formation of gels with varying mechanical properties and topologies. This observation is aligned with previous studies which have found that the stiffness of 3D collagen matrices increases as the concentration of type I collagen added to the matrix is increased [76,77]. A striking result is that by utilizing the 3D Oral Epi-mucosa device, we evaluated the effect of matrix stiffness on the epithelial barrier. We note that apart from stiffness, a variety of other factors, including alterations in matrix porosity, binding kinetics, stress relaxation in the ECM, active contractile forces, and the mechanical remodeling of the matrix and associated feedback between cells and the ECM [78], may also play important roles in barrier function.

The mechanical properties of these gels tune the epithelial barrier function via the localization of E-cadherin to plasma membranes and regulation of mRNA levels of inter-epithelial adhesion molecules. Moreover, substrate stiffness influences ECM organization, particularly the targets responsible for ECM—cell such as focal adhesion molecules (integrin β1) and hemidesmosomes (a6β4 and Lamin 5). Interestingly, our results showed that an increased ratio of ECM stiffness reduced Lamin 5 expression. This is correlated to several studies which have demonstrated the negative effect of high ECM stiffness in hemidesmosome assembly via the Lamin/β4 integrins-dependent binding mechanisms [78,79].

To mimic some of the different fibrillar structures of the native oral ECM [27], we integrated FN into our system and investigated how FN affects barrier function. Several studies have shown that cell behavior is dependent on the ligand density of fibronectin via the EIIIA segment [80,81,82]. To this extent, we investigated the role of FN concentrations at a range of 0 μg/mL to 50 μg/mL in controlling ECM stiffness, which significantly affects cell adhesion, spreading, and migration. The utilization of the 3D Oral Epi-mucosa platform enabled us to capture spatiotemporal changes in epithelial barrier and migration. Specifically, the cells with high FN concentration demonstrated loss of barrier function and high migratory potential, mimicking important aspects of the reepithelization process during wound healing [83].

Our results demonstrated that the presence of FN decreases the formation of adherens junctions via E-cadherin and tight junctions via ZO1 and Occludin. This is in agreement with previous findings, which demonstrate that FN dysregulates cell–cell interactions via E-cadherin and promotes cell–matrix interactions via integrin β1, leading to migratory cellular behavior [82]. Moreover, the negative effect of FN in the hemidesmosome assembly by downregulating the levels of Lamin 5 indicates that cell–cell interactions via E-cadherin may facilitate the hemidesmosome assembly [84].

A deeper understanding of oral diseases requires the investigation of biomechanical cues involved in vivo. The 3D Oral Epi-mucosa device integrates these cues and captures their role in epithelial barrier. The presence of low-level mechanical stress (0.1 kPa) improved epithelial integrity, while changes in ECM stiffness and the presence of FN reversed it. This is linked with the molecules involved in cell–cell and cell–matrix interactions, including E-cadherin and integrins. Although we demonstrated herein the feasibility of investigating the effect of mechanical cues on epithelial barrier, a future challenge may be related to investigate the contribution of other cell populations to matrix properties and epithelial barrier function. Overall, the 3D Oral Epi-mucosa device, as a new experimental model, captures the essential features of a simplified oral sulcus, permits finely tuned cell mechanical challenges, and elucidates new pathways involved in oral diseases.

## Figures and Tables

**Figure 1 bioengineering-10-00517-f001:**
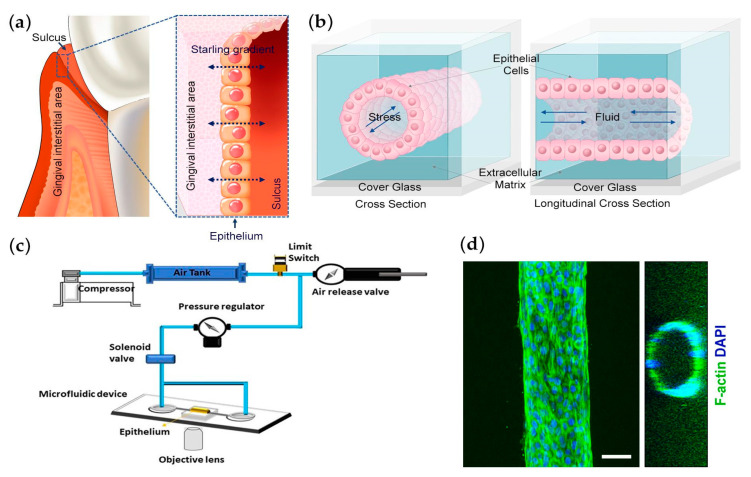
Engineering the 3D Oral Epi-mucosa platform: (**a**) An illustration depicting a starling gradient between the interstitial gingival area and sulcus. (**b**) Schematic of the cross-sectional area of the Epi-mucosa on a chip platform. Oral keratinocytes are seeded in a 3D extracellular matrix within a microfabricated PDMS gasket. (**c**) Schematic of the components of the Oral Epi-mucosa on a chip system. (**d**) Representative confocal immunofluorescence image capturing the formed epithelial cells; HIGKs were stained for F-actin (green) and nuclei for DAPI (blue) (scale bar: 50 μm).

**Figure 2 bioengineering-10-00517-f002:**
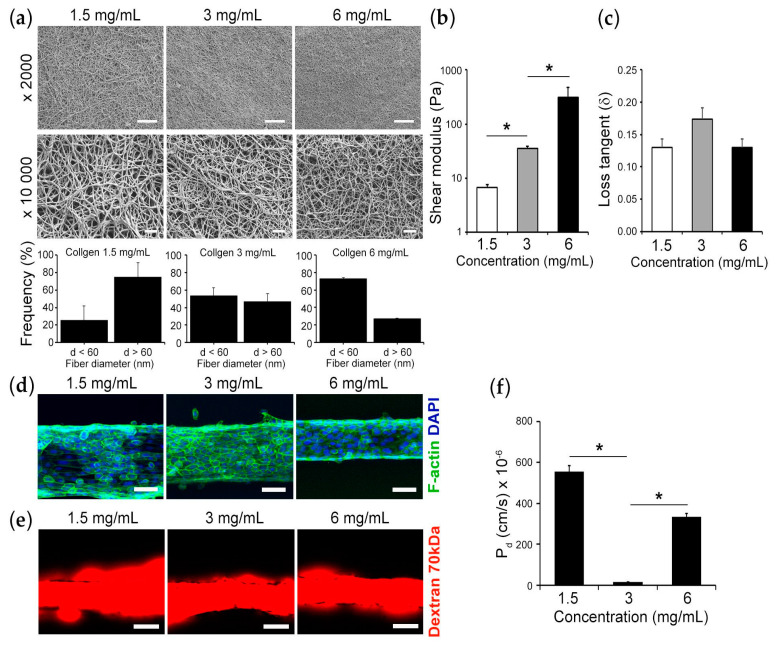
Mechanical properties of 3D ECM controls barrier function: (**a**) Representative SEM images (scale bar: 10 µm for 2000×; 1 μm for 10,000×). Histograms demonstrate the number of fibers with diameter less than 60 nm and more than 60 nm for 1.5 mg/mL, 3 mg/mL, and 6 mg/mL collagen gels. (**b**) Shear stress modulus of 1.5 mg/mL, 3 mg/mL, and 6 mg/mL collagen gels. (**c**) Loss tangent of 1.5 mg/mL, 3 mg/mL, and 6 mg/mL collagen gels. (**d**) Representative confocal microscopy images of cell morphologies on 3D Oral Epi-mucosa platforms of 1.5 mg/mL, 3 mg/mL, and 6 mg/mL collagen gels; HIGKs were stained for F-actin (green) and nuclei for DAPI (blue) (scale bar: 100 µm). (**e**) Representative epithelial integrity images on the devices using fluorescently labeled 70 kDa dextran (scale bar: 100 µm). (**f**) The graph demonstrates the epithelial leakiness by diffusive permeability coefficient (P_d_) for 1.5 mg/mL, 3 mg/mL, and 6 mg/mL collagen gels. Data are expressed as the mean plus or minus one standard deviation and N = 3 at minimum. The asterisk (*****) indicates a *p*-value of *p* < 0.05.

**Figure 3 bioengineering-10-00517-f003:**
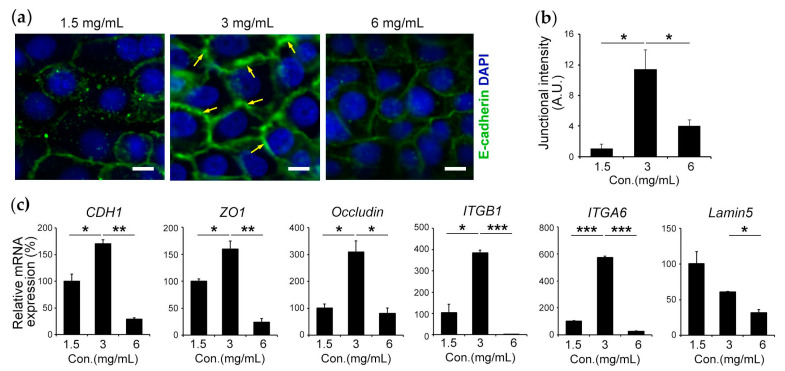
Intercellular interaction is mediated by the mechanical properties of the 3D extracellular matrix: (**a**) HIGKs lying on 1.5 mg/mL, 3 mg/mL, and 6 mg/mL of collagen gels were immunofluorescence stained for E-cadherin (green); examples of membrane localization (yellow arrow) (scale bar: 10 µm). (**b**) Graph demonstrates junctional intensity in arbitrary units (A.U.) of 1.5 mg/mL, 3 mg/mL, and 6 mg/mL of collagen gels; N = 3; n = 5. (**c**) Gene expression analysis of cell–cell adhesion molecules: CDH1, ZO1, Occludin, and cell–matrix contact genes: IΤGβ1, ITGα6, and Lamin5 for cells on 1.5 mg/mL, 3 mg/mL, and 6 mg/mL collagen gels. 18S served as a housekeeping gene. *, *p* < 0.05, **, *p* < 0.01, and ***, *p* < 0.001; N = 3.

**Figure 4 bioengineering-10-00517-f004:**
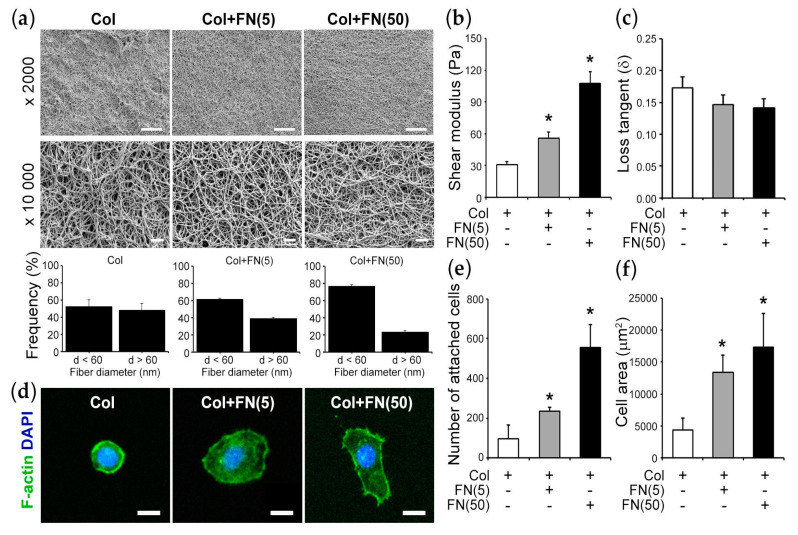
Fibronectin regulates the mechanical properties of the extracellular matrix: (**a**) Representative SEM images (scale bar: 10 µm for 2000×; 1 μm for 10,000×). The histograms represent the number of fibers with less than 60 nm and more than 60 nm diameter for Col, Col + FN(5), and Col + FN(50). (**b**) Shear stress modulus of Col, Col + FN(5), and Col + FN(50) gels. (**c**) Loss tangent of Col, Col + FN(5), and Col + FN(50) gels. (**d**) Representative confocal microscopy images of cell morphologies on Col, Col + FN(5), and Col + FN(50) after 4 h incubation, the cells were stained for F-actin (green) and nuclei for DAPI (blue) (scale bar: 10 µm). (**e**) Number of attached cells per field view on Col, Col + FN(5), and Col + FN(50) gels. (**f**) Cell area analysis of Col, Col + FN(5), and Col + FN(50) gels. *****, *p* < 0.05; N = 3; n ≥ 80.

**Figure 5 bioengineering-10-00517-f005:**
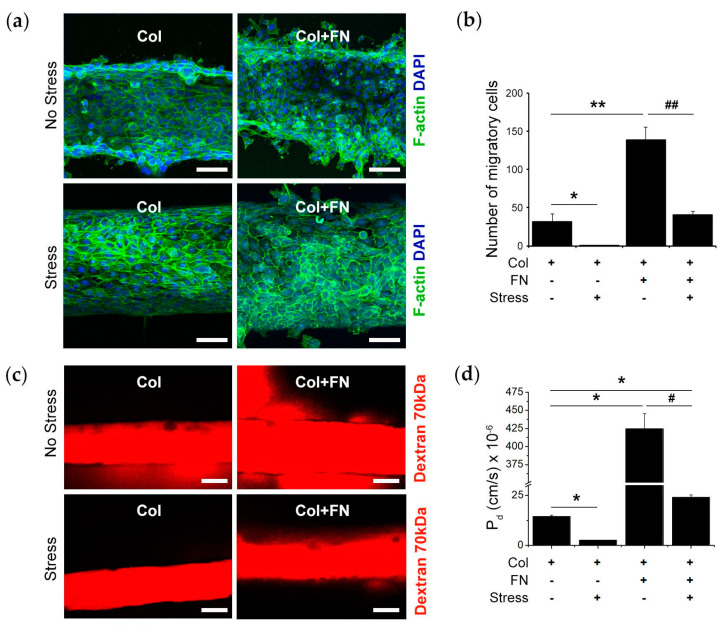
Mechanical stress induces barrier functionality, and FN reverses it: (**a**) Representative confocal microscopy images of cell morphologies on 3D Oral Epi-mucosa platforms in the absence or presence of mechanical stress: Col and Col + FN; HIGKs were stained for F-actin (green) and nuclei for DAPI (blue) (scale bar: 100 µm). (**b**) The graph shows the number of migratory cells in the absence or presence of mechanical stress: Col and Col + FN. (**c**) Representative epithelial integrity images on the devices using fluorescently labeled 70 kDa dextran (scale bar: 100 µm). (**d**) The graph demonstrates the epithelial leakiness by diffusive permeability coefficient (P_d_) for Col and Col + FN in the presence of no stress and stress. *****, *p* < 0.05 Col vs.; ******, *p* < 0.01 Col vs.; #, *p* < 0.05 Col + FN vs. Col + FN with stress; ##, *p* < 0.01 Col + FN vs. Col + FN with stress. N = 3. Col + FN(50) in Figure 4 is denoted as Col + FN in Figure 5.

**Figure 6 bioengineering-10-00517-f006:**
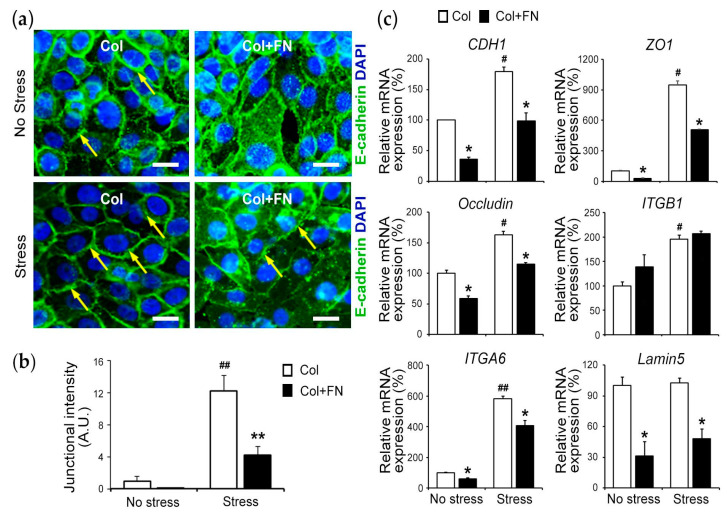
Cell–cell adhesion is mediated by the presence of stress and FN: (**a**) HIGKs at Col and Col + FN under stress and no stress conditions were stained for E-cadherin (green); examples of membrane localization (yellow arrow) (scale bar: 10 µm). (**b**) The graph demonstrates junctional intensity in arbitrary units (A.U.) of Col and Col + FN under stress and no-stress conditions. (**c**) Gene expression profile of intraepithelial markers such as CDH1, ZO1, Occludin, and cell–matrix genes such as IΤGβ1, ITGα6, and Lamin5 in the presence of FN and in no-stress and stress conditions. 18S served as the housekeeping gene. Data are expressed as mean ± SD. *N* = 3. *****, *p* < 0.05 Col vs. Col + FN; ******, *p* < 0.01 Col vs. Col + FN; #, *p* < 0.05 Col vs. Col + FN with stress; ##, *p* < 0.01 Col vs. Col + FN with stress. Col + FN(50) in Figure 4 is denoted as Col + FN in Figure 6.

**Table 1 bioengineering-10-00517-t001:** Different matrix compositions.

	Col	Col + FN(5)	Col + FN (50)
Collagen I (3 mg/mL)	+	+	+
FN (5 µg/mL)	−	+	−
FN (50 µg/mL)	−	−	+

## Data Availability

Not applicable.

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
