# Peer review of "Mechanical Regulation of Oral Epithelial Barrier Function"

_bioengineering, 2023, doi:10.3390/bioengineering10050517_

Round 1

Reviewer 1 Report

This is a review for Bioengineering 2323209 from Stella Alimperti’s group. The goals of this paper was to develop an epithelial tissue 3D model to look at the role of mechanical cues in the barrier function of the oral epithelium. The authors used a microfluidic platform and different concentrations of collagen I and fibronectin to control the mechanical properties of their gels and then cultured oral keratinocytes to look at relevant cell-cell and barrier markers in their system. This is an understudied and important area of research to oral health outcomes. I am supportive of publication of this paper after the following minor comments are addressed. 

Introduction:

I think the introduction would benefit from a sentence or two on the role of FN and COL1 in the oral epithelium or sulcus. These are ubiquitous, I know, but it would help focus non-oral researchers.

Methods:

Silanized, not salinized – page 3

Shear rheometry – Were the sample hydrated or not?

I might have missed it but can the authors include antibody catalog numbers?

70 kDa Texas Red is common but is this kDa particularly relevant to this system?

The Supplemental was clear, thank you.

Results:

Figure 1 is clear, thank you to the authors for this.

I think the Lamin5 PCR is worth more explanation. This shows that the cells are clearly interrogating these gels through a mechanosensitive pathway, obviously, in a conc-dependent way yet we see all the cell-cell and cell-matrix markers peak at 3 mg/mL and not 6 mg/mL. Can the authors comment on this more elsewhere? The reduction of Lamin5 with FN should also be discussed; loss of cell-cell connections in this context might be changing cellular perception of forces not through the matrix but rather through loss of “neighbor” cells.

Discussion:

Gopu Sriram has high quality work on gingival epithelium models, like their recent paper [10.1002/adhm.202202376] and others, that the authors could synergize with in their approach. There is also a lot of work on TRPV4 and epithelium permeability in the sulcus, like  [10.1111/jre.12685] and others. This is only a suggestion and more for the author’s benefit.

This is also somewhat tangential but it occurred to me as I read this. Has anybody looked at any mechanical properties related to periodontitis-infected tissue? 

Author Response

We thank Reviewer #1 for the thoughtful review of the manuscript “The authors used a microfluidic platform and different concentrations of collagen I and fibronectin to control the mechanical properties of their gels and then cultured oral keratinocytes to look at relevant cell-cell and barrier markers in their system. This is an understudied and important area of research to oral health outcomes. I am supportive of the publication of this paper.” Nonetheless, Reviewer #1 raised the following concerns that fell into 6 questions:

Q1: Introduction: I think the introduction would benefit from a sentence or two on the role of FN and COL1 in the oral epithelium or sulcus. These are ubiquitous, I know, but it would help focus non-oral researchers.

Response: As the reviewer pointed out, we described the role of COL1 and FN in the oral epithelium and added to the Introduction (page 2; lines 77-86): “In general, the oral epithelium is lying on a compact collagen fibrous tissue layer, namely the lamina propria, consisting of collagen and fibronectin (FN) [2,27-31]. Normal gingival tissue majorly contains type I collagen and, along with FN provides structural integrity and mechanical strength to the connective tissue [32-35]. Importantly, when gingiva is injured, collagen and FN levels increase and are involved in mechanisms related to inflammatory responses (i.e., cytokines) and activation of immune cells [36,37]. Overall, COL1A1 and FN are essential for maintaining the structure and function of gingiva tissue, and they have been implicated in the regulation of adhesion, wound healing, inflammation, and periodontal tissue regeneration [32-34].”

Q2: (i) Methods: Silanized, not salinized – page 3

(ii) Shear rheometry – Were the sample hydrated or not?

(iii) I might have missed it but can the authors include antibody catalog numbers?

(iv)70 kDa Texas Red is common but is this kDa particularly relevant to this system?

The Supplemental was clear, thank you.

Response: We have edited the Materials and Methods Section based on the comments pointed out by Reviewer #1.

  • We changed “salinized” to “silanized” in line 121 at page 3.
  • For the shear rheometry experiments, the samples were hydrated. We added this information in line 163 at page 4: “First the polymerization temperature was set at 37 °C, and 1 ml of collagen gel solution was loaded into the rheometer. The outer edge of the sample…”
  • We have added the information of E-cadherin antibody on page 5; lines 197-203 as follows: “To capture cell-cell interactions, the cells were incubated with E-Cadherin (24E10) Rabbit mAb (1:100, Cell signaling) overnight at 4 °C. Next day, the cells were washed twice in PBS and a secondary antibody (goat anti-Rabbit IgG (H+L), Alexa Fluor 488 (1:100, ThermoFisher)) was added for 4 h at room temperature. After three time wash, the devices were imaged using a confocal microscope (LSM 800, Carl Zeiss), and image analysis was done using ImageJ [46] by performing a maximum intensity z projection and merging the channels.”
  • Cellular permeability is characterized by better accuracy when utilizing different sizes of macromolecules, which are labeled with fluorophores such as fluorescein isothiocyanate (FITC-dextran), and dextran labeled with rhodamine, etc. The use of 70kDa Texas Red in our system demonstrated a proof of concept regarding the measurement of permeability. This is based on previous in vitro and in vivo studies that have reported the use of 70kDa dextran for measuring oral mucosa barrier (PMID: 35624646) (PMID: 238494) (PMID: 6205133). We added that information in the 2.6 section of Epithelial Permeability Measurement (page 4; lines 180-182) as follows: “To measure the epithelial barrier, as has been demonstrated previously [43-45], fluorescent dextran (70 kDa Texas Red, Thermo Fisher) was introduced into perfusion media (SFM) at a concentration of 12.5 µg/mL.”

Q3: Results: Figure 1 is clear; thank you to the authors for this. I think the Lamin5 PCR is worth more explanation. This shows that the cells are clearly interrogating these gels through a mechanosensitive pathway, obviously, in a conc-dependent way, yet we see all the cell-cell and cell-matrix markers peak at 3 mg/mL and not 6 mg/mL. Can the authors comment on this more elsewhere?

Response: The following was added to explain the correlation between matrix stiffness and lamin5 expression in the Discussion (page 11; lines 428-434) as follows: “Also, substrate stiffness influences ECM organization, particularly the targets responsible for ECM - cell and cell-cell interactions, such as focal adhesion molecules (integrin β1) and hemidesmosomes (a6β4 and Lamin 5). Interestingly, our results showed that an increased ratio of ECM stiffness reduced Lamin 5 expression. This is correlated to several studies which have demonstrated the negative effect of high ECM stiffness in hemidesmosome assembly via the Lamin / β4 integrins dependent binding mechanisms [78,79].”

Q4: The reduction of Lamin5 with FN should also be discussed; loss of cell-cell connections in this context might be changing cellular perception of forces not through the matrix but rather through the loss of “neighbor” cells.

Response:  The following has been added to explain the reduction of Lamin5 with FN (page 12; lines 445-451). “Our results demonstrated that the presence of FN decreases the formation of adheren junctions via E-cadherin and tight junctions via ZO1 and occludin. This is in an agreement with previous findings, which demonstrate that FN dysregulates cell-cell interactions via E-cadherin and promotes cell-matrix interactions via integrin β1 leading to migratory cellular behavior [82]. Also, the negative effect of FN in the hemidesmosome assembly by downregulating the levels of Lamin 5 indicates that cell- cell interactions via E-cadherin may facilitate the hemidesmosome assembly [84].”

Finally, the following has been added to explain the forces mediated cell-cell interactions (page 10; lines 394-395). “Studies have shown that the response of cells to forces is mediated via dependent cell-cell interactions and involves the regulation of cadherins [68-70].”

Q5: Discussion: Gopu Sriram has high quality work on gingival epithelium models, like their recent paper [10.1002/adhm.202202376] and others that the authors could synergize with in their approach. There is also a lot of work on TRPV4 and epithelium permeability in the sulcus, like [10.1111/jre.12685] and others. This is only a suggestion and more for the author’s benefit.

Response: The first paragraph of the Discussion (page 9; lines 361-375) has been changed accordingly, and more papers related to oral-on-a-chip platforms have been reported. Specifically, “A deeper understanding of oral pathophysiology requires the development of 3D culture systems that can recapitulate the oral ECM structures, mechanics, and complex cell–cell and cell-matrix interactions that occur in vivo [51-53]. Recent studies have demonstrated the development of 3D microfluidic platforms that have been widely used in oral, dental, and craniofacial areas [54]. Specific efforts related to oral mucosa include the Gingival Crevice-On-Chip platform, which recapitulates host-oral microbiome interactions [55], oral-mucosa-on a chip has been applied to evaluate the responses of bacteria to dental materials [56], oral mucositis on a chip, which investigates the role of chemo- and radiation treatments in oral tissue [56]. In our study, we developed the 3D Oral Epi-mucosa platform to recapitulate the mechanical forces generated by hydrostatic components and the underlying matrix that the epithelial cells are sensing. The application of this platform permits the measurements of epithelial integrity, avoiding the limitations of animal studies and static micro physiological systems [57]. Also, we were able to investigate key mechanobiological targets involved in epithelial barrier function, which may propagate new treatments for oral diseases, such as periodontitis.” 

Q6: This is also somewhat tangential but it occurred to me as I read this. Has anybody looked at any mechanical properties related to periodontitis-infected tissue?

Response: We have added information in the third paragraph of the Discussion (page 11; lines 409-415) as follows: “There are significant differences in the biomechanics of periodontal tissues in non-periodontitis and periodontitis patients. Several studies have reported that periodontitis-infected tissue has demonstrated a reduction in stiffness and tensile strength and an increase in porosity than healthy tissue due to the destruction of collagen fibers in the periodontal ligament, which is the loading-bearing structures in the periodontium [71-75]. Despite that, much less is known about the oral epithelial barrier in response to the mechanical properties of the matrix.”

Reviewer 2 Report

In this manuscript, the authors reported the effect of ECM stiffness, mechanical stress and ECM protein on oral epithelial barrier functions. In the presence of middle-stiffness ECM (3 mg/mL collagen) and mechanical stress, higher expression levels of adherens and tight junction molecules were reported. The fibronectin was found with a negative regulation roles on epithelial barrier functions. Overall, the manuscript was well prepared and experiments were logically designed. Some issues as below need to be addressed.

1. The authors introduced the significance of mechanical cues in regulation oral tissue homeostasis, it's better to introduce a bit about the previous research related to the current study from the perspectives of both technical and biological aspects.

2. Line 99, for the design of microfluidic platform, please provide the detail dimension and layout for the microfluidic chips. How did the author ensure the mechanical stress applied in the collagen channel without fluid leakage between the medium tubes and the collagen channels?

3. Line 81-84, this section should not be involved in the manuscript.

4. Line 87, please provide the detailed product information or primary cell sorting procedures for the Human Immortalized Gingival Keratinocytes (HIGKs).

5. For Fig 2a, it seems that the collagen concentration is related to both the stiffness and porosity of the ECM. Did the author consider the effects of ECM porosity on epithelial barrier functions? One the other hand, the collagen concentration can also alter the density of ECM proteins present in 3D ECM. Did the author consider these effects?

6. Discussion is not conclusion. Conclusion should be clearly given for this study.  

7. According to the manuscript (results & conclusion), it is surprising to see that the method has no major limitations (disadvantages). Is this correct? If not, please specify at the end of the section.

8. The paper title should be improved with more specificity. The epithelial may be updated with oral epithelial.

9. The references may be updated with more recent research.

Author Response

We thank Reviewer #2 for the comment: Overall, the manuscript was well prepared and experiments were logically designed”

Q1: The authors introduced the significance of mechanical cues in regulation oral tissue homeostasis, it's better to introduce a bit about the previous research related to the current study from the perspectives of both technical and biological aspects.

Response: We added the following to the Introduction section (page 2; lines 71-73), which demonstrates our previous work related to mechanical cues in barrier function as follows: “Specifically, we have developed a 3D microfluidic systems to quantify the mechanical properties of microvascular barrier function in the presence of hydrostatic forces [23,24].”

Q2: Line 99, for the design of microfluidic platform, please provide the detail dimension and layout for the microfluidic chips. How did the author ensure the mechanical stress applied in the collagen channel without fluid leakage between the medium tubes and the collagen channels?

Response: The design of the device and the set up with the pumping system have been demonstrated previously at the publication from Salipante P., Hudson S., Alimperti S. “ Blood vessel-on-a-chip examines the biomechanics of microvasculature” (PMID: 34816867). For further clarification, we have added an extra sentence at the Section 2.3 of the Materials and Methods section (page 3; lines: 113-115):Molds for the microfluidic devices were created with a stereolithography (SLA) 3D printer (Form 3+; Formlabs). Initially, computer-aided design (CAD) models were designed using 3D CAD as demonstrated previously [24].

Q3: Line 81-84, this section should not be involved in the manuscript.

Response: We have moved the information in lines 81-84 to the Acknowledgments section (page 13; lines 486-490).

Q4: Line 87, please provide the detailed product information or primary cell sorting procedures for the Human Immortalized Gingival Keratinocytes (HIGKs).

Response: We have added the following information of HIGKs to the Materials and Methods section (page 2; lines 96-101): “Human immortalized gingival keratinocyte (HIGK) cells were generated by Jeffrey J Mans (University of Florida, Gainesville, FL, USA). The isolation of these cells and immortalization by retrovirus transduction of the HPV type 16, E6/E7 gene was described in [38]. These behave similarly to primary epithelial cultures and have been used extensively to characterize cellular and molecular events that occur in oral mucosal cells [38-41].” 

Q5: For Fig 2a, it seems that the collagen concentration is related to both the stiffness and porosity of the ECM. Did the author consider the effects of ECM porosity on epithelial barrier functions? One the other hand, the collagen concentration can also alter the density of ECM proteins present in 3D ECM. Did the author consider these effects?

Response: We agreed with reviewer’s comment. We have commented in the Discussion (page 11; lines 421-425) as follows: “We note that apart from stiffness a variety of other factors, including alterations in matrix porosity, binding kinetics, stress relaxation in the ECM, active contractile forces, and the mechanical remodeling of the matrix and associated feedback between cells and the ECM [78], may also play important roles in barrier function.

Q6: Discussion is not conclusion. Conclusion should be clearly given for this study. 

Response: The last paragraph of the Discussion (page 12; lines 452-463) has been changed and the conclusion more clearly stated: “A deeper understanding in oral diseases requires the investigation of biomechanical cues involved in vivo. The 3D Oral Epi-mucosa device integrates those cues and captures their role in epithelial barrier. The presence of low-level mechanical stress (0.1 kPa) improved epithelial integrity, while changes in ECM stiffness and the presence of FN reversed it. This is linked with the molecules involved in cell-cell and cell-matrix interactions, including E-cadherin and integrins. Although we demonstrated herein the feasibility of investigating the effect of mechanical cues on epithelial barrier, a future challenge may be related to investigate the contribution of other cell populations to matrix properties and epithelial barrier function. Overall, the 3D Oral Epi-mucosa device as a new experimental model captures the essential features of a simplified oral sulcus and permits finely tuned cell mechanical challenges and elucidate new pathways involved in oral diseases.”

Q7: According to the manuscript (results & conclusion), it is surprising to see that the method has no major limitations (disadvantages). Is this correct? If not, please specify at the end of the section.

Response: Our 3D system integrates key mechanical cues involved in regulation of oral epithelium in vivo. However, how those mechanical cues may contribute to other cell populations of oral mucosa and eventually regulate the epithelial barrier and overall oral homeostasis would be investigated in future studies. Herein, we added a statement at last paragraph of the Discussion (page 12; lines 457-460) as follows: Although we demonstrated herein the feasibility of investigating the effect of mechanical cues on epithelial barrier, a future challenge may be related to investigate the contribution of other cell populations to matrix properties and epithelial barrier function.

Q8: The paper title should be improved with more specificity. The epithelial may be updated with oral epithelial.

Response: We have edited the paper’s title to be more specific as follow: “Mechanical Regulation of Oral Epithelial Barrier Function”

Q9: The references may be updated with more recent research.

Response: We have updated the more recent research by referring efforts on the development of 3D microphysiological systems for oral tissues. This is being reflected in first paragraph of Discussion (page 9; lines: 361-375): “A deeper understanding of oral pathophysiology requires the development of 3D culture systems that can recapitulate the oral ECM structures, mechanics, and complex cell–cell and cell-matrix interactions that occur in vivo [51-53]. Recent studies have demonstrated the development of 3D microfluidic platforms that have been widely used in oral, dental, and craniofacial areas [54]. Specific efforts related to oral mucosa include the Gingival Crevice-On-Chip platform, which recapitulates host-oral microbiome interactions [55], oral-mucosa-on a chip has been applied to evaluate the responses of bacteria to dental materials [56], oral mucositis on a chip, which investigates the role of chemo- and radiation treatments in oral tissue [56]. In our study, we developed the 3D Oral Epi-mucosa platform to recapitulate the mechanical forces generated by hydrostatic components and the underlying matrix that the epithelial cells are sensing. The application of this platform permits the measurements of epithelial integrity, avoiding the limitations of animal studies and static micro physiological systems [57]. Also, we were able to investigate key mechanobiological targets involved in epithelial barrier function, which may propagate new treatments for oral diseases, such as periodontitis.” 

Round 2

Reviewer 2 Report

The authors have addressed all my comments. I have no further comments.